# Genetic Variation in MicroRNA-423 Promotes Proliferation, Migration, Invasion, and Chemoresistance in Breast Cancer Cells

**DOI:** 10.3390/ijms23010380

**Published:** 2021-12-29

**Authors:** Sebastian Morales-Pison, Lilian Jara, Valentina Carrasco, Cristian Gutiérrez-Vera, José Miguel Reyes, Patricio Gonzalez-Hormazabal, Leandro J. Carreño, Julio C. Tapia, Héctor R. Contreras

**Affiliations:** 1Laboratorio de Genética Humana, Programa de Genética Humana, Instituto de Ciencia Biomédicas (ICBM), Facultad de Medicina, Universidad de Chile, Santiago 8380453, Chile; seba.morales.p@gmail.com (S.M.-P.); ljara@med.uchile.cl (L.J.); patriciogonzalez@uchile.cl (P.G.-H.); 2Laboratorio de Biología Estructural y Molecular, Departamento de Biología, Facultad de Ciencias, Universidad de Chile, Santiago 8380453, Chile; valentina.carrasco@gmail.com; 3Millennium Institute on Immunology and Immunotherapy, Programa de Inmunología, Instituto de Ciencias Biomédicas, Facultad de Medicina, Universidad de Chile, Santiago 8380453, Chile; cristian.gutierrez.vera@ug.uchile.cl (C.G.-V.); leandrocarreno@uchile.cl (L.J.C.); 4Clínica Las Condes, Santiago 7591047, Chile; jmreyes@clc.cl; 5Laboratorio de Transformación Celular, Programa de Biología Celular y Molecular, Instituto de Ciencias Biomédicas (ICBM), Facultad de Medicina, Universidad de Chile, Santiago 8380453, Chile; 6Laboratorio de Biología Celular y Molecular, Departamento de Oncología Básico Clínica, Facultad de Medicina, Universidad de Chile, Santiago 8380453, Chile

**Keywords:** breast cancer, microRNA function, microRNA-423, functional study

## Abstract

MicroRNA-423 (miR-423) is highly expressed in breast cancer (BC). Previously, our group showed that the SNP rs6505162:C>A located in the pre-miR-423 was significantly associated with increased familial BC risk in patients with a strong family history of BC. Therefore, in this study, we evaluated the functional role of rs6505162 in mammary tumorigenesis in vitro to corroborate the association of this SNP with BC risk. We found that rs6505162:C>A upregulated expression of both mature miR-423 sequences (3p and 5p). Moreover, pre-miR-423-A enhanced proliferation, and promoted cisplatin resistance in BC cell lines. We also showed that pre-miR-423-A expression decreased cisplatin-induced apoptosis, and increased BC cell migration and invasion. We propose that the rs6505162-A allele promotes miR-423 overexpression, and that the rs6505162-A allele induces BC cell proliferation, viability, chemoresistance, migration, and invasion, and decreases cell apoptosis as a consequence. We suggest that rs6505162:C>A is a functional SNP site with potential utility as a marker for early diagnosis, prognosis, and treatment efficacy monitoring in BRCA1/2-negative BC patients, as well as a possible therapeutic target.

## 1. Introduction

Breast cancer (BC) is the most frequent cancer affecting women worldwide. In Chile, BC has the highest mortality rate among cancers in women (16.6/100,000 women) [1]. Though genetic factors play an important role in etiology, the most well-known susceptibility genes, BRCA1 and BRCA2, only account for about 16% of familial cases [2]. Currently, there is consensus that other susceptibility alleles, called moderate- or low-penetrance genes, could also be responsible for a significant percentage of BC susceptibility [2]. 

In recent years, evidence has emerged to support a role of microRNAs (miRNAs) in BC development and progression [3,4]. These molecules can regulate gene expression by degrading or blocking translation of their specific target mRNAs, mainly by binding to their 3′-unstranslated region (UTR) [5,6]. Approximately 30% of all of human genes are regulated by miRNAs [7,8]. Growing evidence indicates that miRNA variations and mutations are correlated with various human cancers, including BC [9,10,11]. Single-nucleotide polymorphisms (SNPs) are the most common variations in the human genome. SNPs in miRNAs can alter expression, lead to maturation of aberrant miRNA, and affect target binding affinity and specificity [11,12]. Several studies have shown that miRNA-423 (miR-423) plays an important role in tumorigenesis [13,14,15]. In hepatocellular carcinoma, miR-423 promotes cell growth, and regulates G(1)/S transition by targeting p21 Cip1/waf1 [13]. Moreover, in gastric cancers [15,16,17,18], glioblastomas [19,20,21,22], laryngeal carcinomas [14], lung cancers [23,24,25], and endometrial cancers [26,27], miR-423 is implicated in cell proliferation, migration, and invasion. miR-423 also seems to influence proliferation and apoptosis in colorectal [28,29,30,31] and prostate cancers [32].

Zhao et al. [33] suggested that the SNP rs6505162:C>A in pre-miR-423 affects mature miRNA expression, and that miR-423 plays a potentially oncogenic role in breast tumorigenesis. Our group previously evaluated the association between rs6505162:C>A and BC susceptibility in a Chilean population using a case-control study. We concluded that the rs6505162-A allele was significantly associated with increased familial BC risk in patients with a strong family history (OR = 1.7 [95% CI 1.0–2.0] *p* = 0.05) [34]. Given these promising findings, the goal of the present study was to perform a functional characterization of the rs6505162:C>A polymorphism, evaluating its role in several breast tumorigenesis processes in vitro.

## 2. Results

### 2.1. miR-423 Expression Was Upregulated by Rs6505162:C>A

In a previous article, our findings supported the contention that allele A (the minor allele, or MA) of rs6505162:C>A in pre-miR-423 increased familial BC risk. Consequently, we were interested in exploring the effect of pre-miR-423 rs6505162:C>A on mature miR-423 expression. Pre-miR-423 produces two mature sequences: miR-423-3p and miR-423-5p. We constructed pre-miR-423-C (wild-type) and pre-miR-423-A (MA) expression vectors, which were transfected into MCF-10A, MCF-7, and MDA-MB-231 cells to produce stable populations. Expression was evaluated by RT-qPCR. Figure 1A shows that miR-423-3p and miR-423-5p expression levels were very low, and were similar in MCF-10A cells regardless of transfection with pre-miR-423-C or pre-miR-423-A. On the other hand, miR-423-3p and miR-423-5p levels were significantly increased in both MCF-7 and MDA-MB-231 BC cells transfected with pre-miR-423-A versus pre-miR-423-C (Figure 1B,C). These results suggest that the expression levels of both mature miR-423 sequences (3p and 5p) are upregulated in BC cells by the presence of the rs6505162-A allele in pre-miR-423. 

### 2.2. Pre-miR-423-A Enhanced Breast Cancer Cell Proliferation

MCF-7 and MDA-MB-231 transfected with pre-miR-423-A, pre-miR-423-C, or empty vector were studied to assess the effect of pre-miR-423 rs6505162:C>A. Proliferation was evaluated in both stable cell lines with CFSE-based proliferation assays at 24 and 48 h. Figure 2A shows that transfection with pre-miR-423-C had no effect on the proliferation index in MCF-7 cells compared to transfection with empty vector, after either 24 or 48 h. However, proliferation indices were significantly higher (*p* < 0.01) in cells transfected with pre-miR-423-A as compared to pre-miR-423-C or empty vector at both 24 and 48 h (Figure 2A). Figure 2B shows the results for MDA-MB-231 cells. No significant differences in the proliferation index were observed among cells transfected with pre-miR-423-A, pre-miR-423-C, or empty vector at 24 h. However, cells transfected with pre-miR-423-A had significantly higher proliferation indices than those transfected with pre-miR-423-C or empty vector at 48 h (*p* < 0.001) (Figure 2B). Furthermore, we evaluated the proliferation index in the normal breast cell line MCF-10A, where we observed no significant differences when cells were transfected with pre-miR-423-C, pre-miR-423-A, or empty vector, at either 24 or 48 h (Figure 2C). The results obtained indicate that pre-miR-423-A significantly enhanced proliferation of both luminal A and triple-negative representative BC cells.

### 2.3. Pre-miR-423-A Expression Promoted Cisplatin Resistance in Breast Cancer Cells

It is very well known that some miRNAs can regulate resistance to antineoplastic drugs [35]. To study the effects of the SNP rs6505162:C>A on cisplatin sensitivity, stable MCF-7 and MDA-MB-231 BC cells were cultured with various concentrations of cisplatin (0, 40, and 80 µM) for 48 h. An MTS assay was used to evaluate cisplatin sensitivity. Survival in the presence of cisplatin was significantly higher for MCF-7 cells transfected with pre-miR-423-A than for those transfected with pre-miRNA-423-C or empty vector (Figure 3A). Similar results were observed for stable MDA-MB-231 cells (Figure 3B). In the triple-negative BC cell line, a slight, but significant, cisplatin resistance was observed in cells transfected with pre-miR-423-C versus empty vector. In summary, the results support the assertion that the SNP rs6505162-A is a risk allele, as the presence of allele A in pre-miR-423 was associated with a significant elevation in cell survival and cisplatin resistance as compared to allele C or empty vector for two concentrations of cisplatin. 

### 2.4. Pre-miR-423-A Expression Decreased Cisplatin-Induced Apoptosis of Breast Cancer Cells

To examine the role of the SNP rs6505162:C>A in cisplatin-induced apoptosis, MCF-7 and MDA-MB-231 cells transfected with pre-miR-423-C, pre-miR-423-A, or empty vector were treated with 100 µM cisplatin. Apoptosis was estimated by measuring caspase 3/7 activity. Figure 4A,B shows significantly lower levels of cisplatin-induced apoptosis at 48 h in MCF-7 and MDA-MB-231 cells transfected with pre-miR-423-A compared to those transfected with pre-miR-423-C (*p* < 0.0001) or empty vector (*p* < 0.0001). These results demonstrate that miR-423 upregulation, attributable to the presence of the rs6505162-A allele in pre-miR-423, inhibited cisplatin-induced apoptosis by decreasing caspase 3/7 activity. The results are consistent with our previously-published findings implicating rs6505162-A as a risk allele.

### 2.5. Pre-miR-423-A Expression Increased Migration and Invasion in Breast Cancer Cells

To evaluate the effect of rs6505162:C>A on migration and invasion in BC cells, migration and invasion assays were performed using MCF-7 and MDA-MB-231 cells transfected with pre-miR-423-C, pre-miR-423-A, or empty vector. The cells were plated in Transwell chambers with or without Matrigel. Cell migration was significantly greater in MCF-7 cells transfected with pre-miR-423-A versus pre-miR-423-C (*p* < 0.01) or empty vector (*p* < 0.01) (Figure 5A). Similar results were observed for stable MDA-MB-231 cells (Figure 5B). Our results also demonstrated increased cell invasion in both MCF-7 (Figure 6A) and MDA-MB-231 (Figure 6B) cells transfected with pre-miR-423-A. No significant differences in migration or invasion were observed between MCF-7 and MDA-MB-231 cells transfected with pre-miR-423-C versus empty vector (Figure 5A,B and Figure 6A,B). Therefore, it was demonstrated that miR-423 upregulation in the presence of the pre-miR-423 rs6505162-A allele enhanced migration and invasion of luminal A and triple-negative BC cells. 

## 3. Discussion

It is well established that miRNAs participate in the initial and developmental stages of numerous types of cancers [36,37]. Notably, miRNAs are frequently located in cancer-associated genome regions [38], and seem to regulate almost all cancer-associated genes [39]. Mutations or SNPs located in pri- or pre-miRNAs may affect the transcription of miRNA primary transcripts, processing of miRNA precursors to mature miRNAs, and miRNA-mRNA interactions [40].

Numerous studies have indicated that miR-423 shows distinct expression patterns and functions in cellular processes, such as cell proliferation, apoptosis, tumor metastasis, and chemoresistance [41]. miR-423 is highly expressed in multiple cancer types, including hepatocellular carcinomas, head and neck cancers, gastric cancers, colorectal cancers, glioblastomas, lung cancers, ovarian cancers, laryngeal carcinomas, endometrial cancers, prostate carcinomas, and breast cancers [15,41,42,43,44,45]. A pre-miRNA usually produces two mature sequences: miRNA-5p and miRNA-3p. It is thought that one pre-miRNA is typically processed into one mature miRNA, which incorporates into the RISC complex, whereas the other arm of the pre-miRNA degrades rapidly. However, pre-miR-423 can simultaneously produce two mature sequences, possibly as a consequence of structural characteristics that allow both strands to join the RISC complex [13]. It has been reported that both of the mature miR-423 sequences, miR-423-3p and miR-423-5p, are involved in tumorigenesis [33]. In laryngeal carcinoma, miR-423-3p increases cell proliferation, invasion, and migration via modulation of AdipoR2 [14]. In gastric cancer cells, miR-423-5p regulates cell proliferation and invasion by targeting trefoil factor 1 [15], and in hepatocellular carcinoma, miR-423-3p overexpression promotes cell growth and regulates G1/S transition by targeting *p21Cip/WaF1*, a tumor suppressor gene [13]. Overexpression of miR-423-5p induces upregulation of *p*-ERK1/2 and *p*-AKT, and enhances glioma cell proliferation, as well as angiogenesis and metastasis, by targeting inhibitor of growth 4 (ING-4) [19]. miR-423-3p acts as an oncogene, and promotes cell proliferation, migration, and invasion in lung cancer [24], and in endometrial cancer, miR-423 promotes proliferation, migration, invasion, and chemoresistance [26]. In prostate cancer, miR-423-5p enhances proliferation and decreases apoptosis [32,41]. miR-423-5p has a protective effect in colorectal cancer, downregulating cell proliferation, and upregulating apoptosis [28,41], and in ovarian cancer, miR-423-5p decreases cell proliferation and invasion [41,46]. In breast cancer, Zhao et al. [33] showed that both forms of mature miR-423 were expressed in nine breast cancer cell lines. In addition, Xia et al. [47] reported that miR-423 acts as an oncogene to promote tumor cell proliferation and migration, inhibiting ZFP36 expression via the Wnt/β-catenin signaling pathway of BC cells. Furthermore, Dai et al. (2020) [48] demonstrated that miR-423 upregulation enhances BC cell invasion through the NF-κβ signaling pathway. 

It has been established that SNPs play a crucial role in the development of human tumors [49,50]. A meta-analysis of miR-423 polymorphisms and cancer prognoses suggests that rs6505162 is a prognostic marker in all common human cancers [41]. SNP rs6505162:C>A, mapping to chromosome 17 at 17q11.2, is located in the pre-miRNA region of miR-423, 12bp from the 3′ end of mature miR-423 [33]. In a previous study, our group showed that the rs6505162 allele A (rs6505162-A) is significantly associated with familial BC risk in patients with a strong family history [34]. Therefore, in this study, we performed an in vitro evaluation of the role of the SNP rs6505162:C>A in several breast tumorigenesis processes. Our results showed expression of both miR-423-3p and miR-423-5p in the MCF-7 and MDA-MB-231 BC cell lines, consistent with results from Zhao et al. [33]. We also observed expression of both mature miRNAs (5p and 3p) in the normal breast cell line, but at very low levels. Moreover, we found significantly increased levels of mature miR-423-3p and miR-423-5p in cancer cell lines transfected with pre-miR-423-A as compared to those transfected with pre-miR-423-C (the wild-type allele) in both MCF-7 and triple-negative BC cell lines, but not in the normal epithelial breast cancer line. Our results indicate that the rs6505162-A allele affects expression of both mature sequences, miR-423-3p and -5p, in cancer cell lines. Our findings are consistent with those of Zhao et al. [33]. We also evaluated the effect of rs6505162:C>A on cell proliferation using flow cytometry, as this technique is the most sensitive method available for assessing the proliferation index. Our results showed that the presence of rs6505162-A increased the proliferation rate of the BC cell line MCF7 and triple-negative BC cell line MDA-MB-231. It is likely that upregulation of miR-423 expression levels as a consequence of the presence of the rs6505162-A allele enhances BC cell proliferation. Our results are the first to report a functional characterization of rs6505162:C>A.

Cisplatin is a well-known and effective anticancer drug that has been used to treat numerous types of malignant tumors [51]. However, many tumor cells show intrinsic or acquired resistance to chemotherapeutic drugs [52,53]. miRNAs have been demonstrated to serve essential roles in chemotherapy sensitivity [54]. Yang et al. [55] reported that miR-214 promotes cell survival, and induces cisplatin resistance by targeting PTEN in ovarian cancer. Kong et al. [56] demonstrated that miR-155 overexpression promotes BT-474 breast cells resistant to paclitaxel, VP16, and doxorubicin, whereas miR-155 downregulation sensitizes HS578T cells to these drugs. Yu et al. [57] reported that miR-17/20 overexpression increases doxorubicin-induced apoptosis in MCF-7 breast cancer cells by targeting AKT1. The present study demonstrated that miR-423 overexpression as a consequence of the presence of the rs6505162-A allele in pre-miR-423 increased survival of BC cells following cisplatin treatment. These results suggest that miR-423 induces drug resistance in BC cells, consistent with the report from Xia et al. [47]. 

Inhibition of apoptosis can promote tumor progression, whereas induction of apoptosis can inhibit tumor progression [58]. Li et al. [26] showed that miR-423 inhibits cisplatin-induced apoptosis by modulating caspase 3/7 and Bcl-2 expression levels in endometrial cancer cells. In glioma cells, miR-423-3p inhibition leads to an induction of apoptosis [20]. Our study results demonstrated that in BC cells, miR-423 upregulation as a consequence of the rs6505162-A allele inhibited cisplatin-induced apoptosis by decreasing caspase 3/7 activity, as has been observed in other cancers.

Metastasis is the leading cause of cancer mortality. Cell invasion and migration represent the initial steps of metastasis [59]. miR-423 is a common marker of cross-cancer metastasis in metastatic samples [41]. miR-423 overexpression in endometrial cancer promotes metastasis of cancer cells [26,27]. miR-423-5p participates in gastric cancer cell invasion [17]. miR-423-3p affects tumor cell metastasis in laryngeal and lung cancers [14,24,25]. In addition, miR-423-5p promotes glioma cell invasion [19]. Thus, the literature suggests that abnormal expression of miR-423 can influence cancer cell metastasis in human tumors. On the other hand, only Mir et al. [60] have evaluated the correlation between the SNP rs6505162:C>A and the metastatic process in BC patients. The authors showed that rs6505162 allele A had a strong and significant association (*p* < 0.009) with metastasis in Saudi Arabian BC patients, indicating that this allele is associated with disease progression and distant metastasis status. Our results show that the rs6505162 A allele in pre-miR-423 increased migration and invasion in BC cells as a consequence of miR-423 overexpression. Moreover, we carried out an in-silico analysis using the miRWalk database (http://mirwalk.umm.uni-heidelberg.de/ (accessed on 20 December 2021)), finding that TNIP2 is a possible target of miR-423. Huber et al. [61] showed that the IKK-2/IkappaBalpha/NF-kappaB pathway is required for induction and maintenance of epithelial–mesenchymal transition (EMT), which is a crucial step in the initiation of metastatic process. The authors found that inhibiting NF-kappaB signaling prevented EMT in Ras-transformed epithelial cells, whereas activation of this pathway promoted the transition to a mesenchymal phenotype. Furthermore, inhibition of NF-kappaB activity in mesenchymal cells caused a reversal of EMT, suggesting that NF-kappaB is essential for both the induction and maintenance of EMT. In accordance with this previous report, Dai et al. [48] observed that upregulation of miR-423 led to activation of the NF-kappaB signaling pathway by repressing TNIP2 expression, suggesting that miR-423 is a crucial factor that enhances breast cancer cell invasion. Therefore, miR-423 overexpression as a consequence of the presence of the rs6505162 allele A could trigger an initiation of the metastatic process by activating the NF-kappaB signaling pathway. This finding highlights this miRNA as a promising prognostic and therapeutic marker for metastatic BC. 

## 4. Materials and Methods

### 4.1. Cell Lines

Normal human breast epithelial cell lines (MCF-10A) and BC cell lines (MCF-7 and MDA-MB-231) were purchased from American Type Culture Collection (Manassas, VA, USA). The BC cell lines were selected because of their low miR423-5p and miR423-3p expression levels [33,62]. MCF-10A is a not tumorigenic cell line, is ER (−), and is the most commonly-used normal breast cell model. MCF-7 is ER (+), PR (+), and HER2 (−), and, therefore, can be classified as luminal A. MDA-MB-231 is a triple-negative cell line. Both BC cell lines are negative for BRCA1 and BRCA2 mutations. MCF-10A cells were grown in Dulbecco’s Modified Eagle Medium F-12 (DMEM/F-12) supplemented with 5% horse serum (Gibco; Thermo Fisher Scientific, Waltham, MA, USA). MCF-7 and MDA-MB-231 were seeded in high-glucose DMEM containing 10% fetal bovine serum (FBS Gibco; Thermo Fisher Scientific, Waltham, MA, USA). The cells were cultured at 37°C in a humidified atmosphere containing 5% CO_2_. After reaching 70–80% confluence, cells were harvested for use in further experiments.

### 4.2. miR-423 Expression Vector Construction and Cell Transfection

We designed one pair of primers (forward: 5′-CCGAAGTTTGAGGGAGAAACT-3′ and reverse: 5′-TTCCTGGCTTCCTTAGAGGG-3′) to amplify pre-miR-423 rs6505162-C and pre-miR-423 rs6505162-A sequences from genomic DNA extracted from CC and AA homozygous patients for rs6505162. PCR products were cut from agarose gel, purified, and inserted into pcDNA3.3 TOPO-TA (Invitrogen; Thermo Fisher Scientific, Waltham, MA, USA). Correct insertion of pre-miR-423-A and pre-miR-423-C into the expression vector was further confirmed by SANGER sequencing. Lipofectamine 2000 (Invitrogen; Thermo Fisher Scientific, Waltham, MA, USA) was used to transfect the recombinant vectors into MCF-10A, MCF-7, and MDA-MB-231 cells. After selection using G418 (800 ug/mL) (Sigma Aldrich, St. Louis, MO, USA), three groups of stable cell populations transfected with pre-miR-423-A, pre-miR-423-C, or empty vector were obtained. miR-423 expression was detected via reverse transcription-quantitative polymerase chain reaction (RT-qPCR).

### 4.3. Analysis of miR-423-5p and miR-423-3p Expression by RT-qPCR

Total small RNA was extracted from the transfected BC cells using the mirVana kit (Invitrogen; Thermo Fisher Scientific, Waltham, MA, USA) according to manufacturer protocol. RNA concentration was determined using the NanoDrop system. Small RNA was reverse transcribed to complementary DNA (cDNA) using the TaqMan microRNA Reverse-Transcription Kit (Applied Biosystems, Foster City, CA, USA). miR-423-5p and miR-423-3p expression levels were determined using specific TaqMan microRNA assays (Applied Biosystems, Foster City, CA, USA) in a StepOne Plus real-time PCR system (Applied Biosystems, Foster City, CA, USA). qPCR was performed as follows: 95 °C for 10 min, followed by 40 cycles of 95 °C for 15 s, and 60 °C for 60 s. All reactions were performed in triplicate. U6 (Applied Biosystems, Foster City, CA, USA) was used as an endogenous control to calculate miR-423 expression in BC cells. miRNA expression levels were measured based on threshold cycle (Ct), and relative expression levels were calculated using the 2^−ΔΔCt^ method [63].

### 4.4. Cell Proliferation Assay

To assess the effect of pre-miR-423 rs6505162:C>A on BC cell proliferation, CFSE assay (BioLegend Way, San Diego, CA, USA) was performed according to manufacturer protocol. The MCF-7 and MDA-MB-231 transfected cells were resuspended and collected at a density of 1 × 10^6^ cells/well. Cells were then labelled with 5 µM CFSE dye (CFSE Cell Division Tracker Kit) for 30 min. Staining was halted by adding 5 volumes of ice-cold culture media. BC cells were subsequently cultured at 37 °C in a humidified atmosphere with 5% CO_2_ for 24 and 48 h. Fluorescence was measured in a BD FACSVerseTM flow cytometer (BD Biosciences, San Jose, CA, USA). The proliferation index, analyzed using FlowJo V.10.1 software (BD Biosciences, San Jose, CA, USA), was calculated based on the sum of cells in all generations divided by the calculated number of original parent cells. All experiments were performed in triplicate.

### 4.5. Cell Viability Assay 

To determine the effects of pre-miR-423-C and pre-miR-423-A on BC cell viability, an MTS^®^ assay (Promega Corporation, Madison, WI, USA) was performed according to manufacturer protocol. Stable BC clones of MCF-7 and MDA-MB-231 cells transfected with pre-miR-423-A, pre-miR-423-C, or empty vector were seeded in 96-well plates at a density of 5 × 10^4^ cells. After 24 h incubation, various concentrations of cisplatin (0, 40, and 80 µM) were added. BC cells were subsequently cultured at 37 °C in a humidified atmosphere containing 5% CO_2_ for 48 h. MTS was added to each well, and incubated for 4 h. Absorbance was measured at 490 nm on a multiplate reader (BioTek Instrument, Inc., Winooski, VT, USA). All experiments were performed three times in triplicate.

### 4.6. Apoptosis Analysis

To examine the effects of pre-miR-423-A and pre-miR-423-C on cisplatin-induced apoptosis of BC cells, caspase 3/7 activity was assessed according to manufacturer protocol (Caspase-Glo 3/7 assay systems; Promega Corporation, Madison, WI, USA). Stable BC clones of MCF-7 and MDA-MB-231 cells transfected with pre-miR-423-A, pre-miR-423-C, or empty vector were seeded in 96-well plates at a density of 1 × 10^4^ cells. The cells were cultured in DMEM overnight at 37 °C in a humidified incubator. Subsequently, the culture medium was replaced with a medium containing 0 or 100 μg/mL cisplatin. Cells were cultured for a further 48 h. Caspase-Glo reagent (Promega Corporation) was added to each well, and incubated for 3 h at room temperature with gentle agitation. Caspase 3/7 activity was measured on a BioTek Synergy HTX Multi-Mode reader (BioTek Instrument, Inc., Winooski, VT, USA). All experiments were performed three times in triplicate.

### 4.7. Migration and Invasion Assay

Transwell migration and invasion assays were performed to determine the impact of pre-miR-423 rs6505162:C>A on cell migration and invasion in BC cells. For the invasion assay, the upper chambers of Transwell plates were coated with 100 μL Matrigel (Sigma-Aldrich, St. Louis, MO, USA). MCF-7 and MDA-MB-231 cells transfected with pre-miR-423-C and pre-miR-423-A were suspended in culture medium without FBS, and plated in the upper chamber at a density of 1 × 10^5^/mL. Culture medium containing 10% FBS was added to the lower chamber. The cells were incubated for 12 to 16 h at 37 °C in a humidified atmosphere containing 5% CO_2_. Remaining cells in the upper chamber were removed with a cotton swab, and cells in the lower chamber were stained at room temperature for 60 min with 0.1% crystal violet plus 2% methanol. The invaded cells were counted under light microscopy in ten randomly-selected fields (magnification, ×40). BC cells transfected with empty vector were used as controls. All experiments were performed in triplicate.

### 4.8. Statistical Analysis

Statistical analyses were performed using GraphPad Prism software (GraphPad Software, La Jolla, CA, USA). Data are presented as mean ± SD, and represented graphically with box plots. One- or two-way ANOVA was used to analyze for significant differences among groups. *p*-values < 0.05 were considered to indicate a statistically significant difference.

## 5. Conclusions

We carried out the first functional characterization of the rs6505162 polymorphism. We propose that the rs6505162-A allele promotes miR-423 overexpression. Moreover, the rs6505162-A allele induces BC cell proliferation, viability, chemoresistance, migration, and invasion, as well as decreasing cell apoptosis, as a consequence of this miR-423 overexpression. The current results confirm our previous findings indicating that the rs6505162-A allele is a risk allele for BC. Taken together, the present results indicate that the rs6505162-A allele upregulates miR-423, and, consequently, plays an oncogenic role in BC tumorigenesis and likely other cancers. This SNP is a potential marker in BRCA1/2-negative BC. The variant may also a useful biomarker for early diagnosis, prognosis, and treatment efficacy monitoring, as well as a possible therapeutic target.

## Figures and Tables

**Figure 1 ijms-23-00380-f001:**
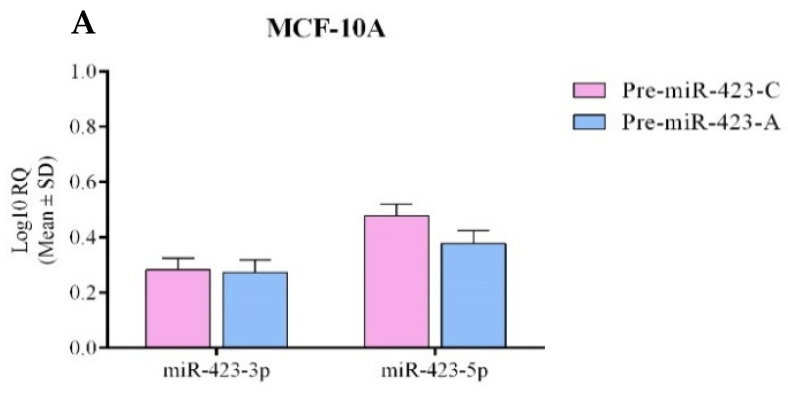
Mature miR-423 expression was differentially regulated by the SNP rs6505162:C>A in normal and cancerous breast cells. Levels of mature miR-423-3p and miR-423-5p were measured in stable populations of normal breast MCF-10A (**A**), MCF-7 (**B**), and MDA-MB-231 (**C**) BC cell lines, which were transfected with expression vectors containing pre-miR-423-A or pre-miR-423-C. Fold changes were normalized to the empty vector. Data represent means (±SD) of three independent experiments (* *p* < 0.05; ** *p* < 0.01; *** *p* < 0.001).

**Figure 2 ijms-23-00380-f002:**
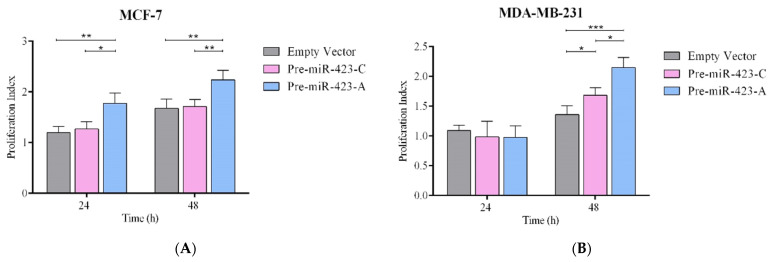
Cell proliferation was increased by the SNPs rs6505162:C>A in luminal A and triple-negative BC cells. Proliferation index was measured by CFSE-based flow cytometry in stable populations of MCF-7 luminal A (**A**), MDA-MB-231 triple-negative (**B**), and MCF-10A (**C**) cells transfected with expression vectors containing pre-miR-423-A, pre-miR-423-C, or empty vector at 24 and 48 h. Data represent means (±SD) of three independent experiments (* *p* < 0.05; ** *p* < 0.01; *** *p* < 0.001).

**Figure 3 ijms-23-00380-f003:**
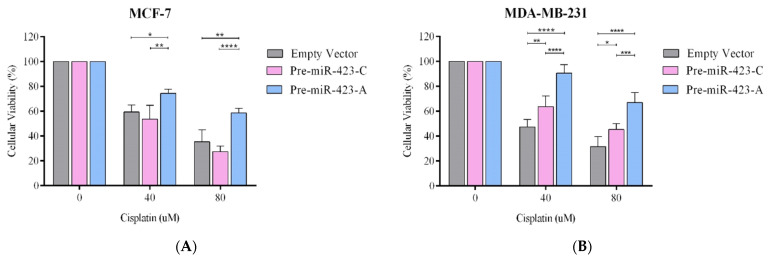
Resistance to cisplatin was differentially modulated by the SNP rs6505162:C>A in luminal A and triple-negative BC cell lines. Resistance to cisplatin was measured by MTS assay in stable populations of MCF-7 luminal A (**A**) and MDA-MB-231 triple-negative (**B**) BC cells transfected with vectors containing pre-miR-423-A, pre-miR-423-C, or empty vector. Data represent means (± SD) of three independent experiments (* *p* < 0.05; ** *p* < 0.01; *** *p* < 0.001; **** *p*< 0.0001).

**Figure 4 ijms-23-00380-f004:**
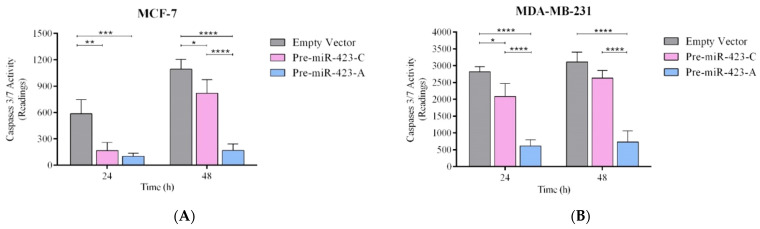
Pre-miR-423-A expression protected against apoptosis in luminal A and triple-negative BC cell lines. Apoptosis was assessed by measuring caspase 3/7 activity in stable populations of MCF-7 luminal A (**A**) and MDA-MB-231 triple-negative (**B**) BC cells transfected with vectors containing pre-miR-423-A, pre-miR-423-C, or empty vector. Data represent means (±SD) of three independent experiments (* *p* < 0.05; ** *p* < 0.01; *** *p* < 0.001; **** *p* < 0.0001).

**Figure 5 ijms-23-00380-f005:**
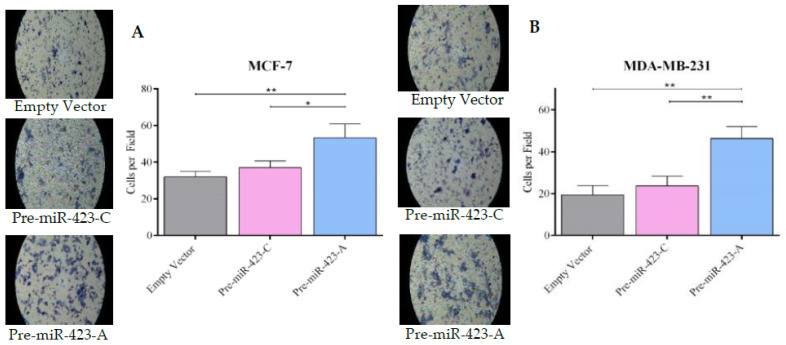
SNP rs6505162:C>A expression increased cell migration in luminal A and triple-negative BC cell lines. Cell migration was measured in stable populations of MCF-7 luminal A (**A**) and MDA-MB-231 triple-negative (**B**) BC cells transfected with expression vectors containing pre-miR-423-A, pre-miR-423-C, or empty vector. The number of migrated cells was quantified by counting the cells in the pictures taken. Data represent means (±SD) of three independent experiments (* *p* < 0.05; ** *p* < 0.01).

**Figure 6 ijms-23-00380-f006:**
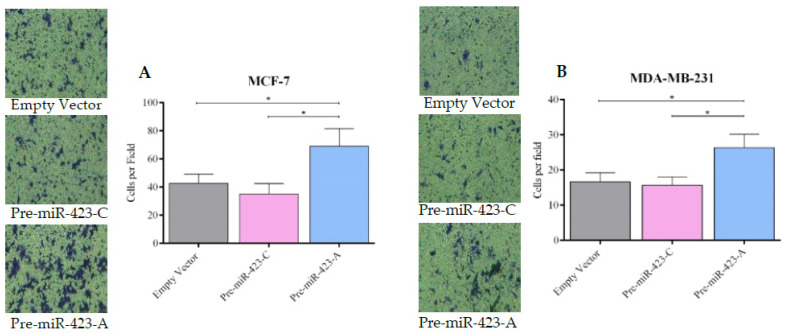
SNP rs6505162:C>A expression increased cell invasion in luminal A and triple-negative BC cell lines. Cell invasion was measured in stable populations of MCF-7 luminal A (**A**) and MDA-MB-231 triple-negative (**B**) BC cells transfected with expression vectors containing pre-miR-423-A, pre-miR-423-C, or empty vector. The number of invasive cells was quantified by counting the cells in the pictures taken. Data represent means (±SD) of three independent experiments (* *p* < 0.05).

## Data Availability

All data are shown within the manuscript.

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
