# Peer review of "Genetic Variation in MicroRNA-423 Promotes Proliferation, Migration, Invasion, and Chemoresistance in Breast Cancer Cells"

_ijms, 2021, doi:10.3390/ijms23010380_

Round 1

Reviewer 1 Report

In the paper entitled “Genetic variation in microRNA-423 promotes proliferation, migration, invasion, and chemoresistance in breast cancer cells” Morales-Pison S et al report the functional characterization of the SNP rs6505162:C>A polymorphism to evaluate its role in breast tumorigenesis by performing in vitro tests in luminal A and triple negative breast cancer cell lines.

This work is based on the previous group finding showing that the minor A allele of rs6505162:C>A in pre-miR-423 increased familial BC risk, in a Chilean population using a case-control study. Moreover, others groups have suggested that the SNP rs6505162:C>A in pre-miR-423 affects mature miRNA expression and that miR-423 plays a potentially oncogenic role in tumorigenesis. Consequently, the authors have investigated the effect of pre-miR-423 rs6505162:C>A on mature miR-423-5p and -3p expression and on their potential oncogenic role in tumorigenesis. They have shown that the stable overexpression of pre-miR-423-A enhances the expression of both mature forms of miR-423, increases proliferation, migration and invasion and promotes cisplatin resistance in MCF7 and MDA-MB-231. The results support the conclusion that the SNP rs6505162-A is a risk allele, as the presence of allele A in pre-miR-423 is associated with a significant increase in breast cancer cell proliferation, survival and cisplatin resistance, in accord with previous works.

The paper is well presented, the methods are well described and the suggestion that this SNP could be a potential marker in BRCA1/2-negative breast cancer seems reasonable.

In 4.3). Analysis of miR-423-5p and miR-423-3p expression by RT-qPCR; which reverse transcription and TaqMan qPCR kit have been used?  The fast advanced with 3’ polyA and 5’ extensions or the RT specific Kit?

It seems that the SNP by enhancing miR-423 expression and activity not only increases early events in tumor formation but would also be involved in tumor cell dissemination. Is there any meta-analysis to support the association of the SNP and metastatic breast cancer? Moreover, the same effects in in vitro tests were reported in luminal A (MCF7) and triple negative (MDA-MB-231) cell lines, suggesting a common gene targeting in both type of breast cancer. The authors should report in-silico analysis of these targeted genes to decipher the molecular mechanism underlying the miR-423 oncogenic activity, especially in invasive process.

In the abstract the authors report the methods used in the study. It should be better to clearly present the SNP rs6505162:C>A polymorphism and consequences on miR-423 expression and potentially in clinic.

Author Response

Regarding the comments of reviewer 1:

a) In relation to point 4.3, we used a RT-specific TaqMan qPCR kit as detailed in Materials and methods, line 336.

b) Regarding the existence of association studies of the SNP rs6505162 with the process of metastasis in breast cancer, after reviewing the literature we found only one article, in patients with breast cancer from Saudi Arabia. In the discussion we add the information provided by this article: “On the other hand, the study of Mir et al. [60] is the only one to has evaluate the correlation between the SNP rs6505162:C>A and the metastatic process in BC patients. The authors show that the allele A of rs6505162 had a strong significant association (p<0.009) in Saudi Arabian BC patients with metastasis, indicating that this allele is associated with the disease progression and distant metastasis status” (lines 285-288).

c) An in-silico analysis was performed to search for possible targets of miR-423 involved in the metastatic process. The result of this analysis made it possible to identify the TNIP2 gene as a possible target for miR-423. Therefore, the following paragraph was added to the Discussion, lines 290-301: “Moreover, we carried out an in-silico analysis using the miRWalk database (http://mirwalk.umm.uni-heidelberg.de/), finding that TNIP2 is a possible target of miR-423. Huber et al. [61] showed that the IKK-2/IkappaBalpha/NF-kappaB pathway is required for induction and maintenance of epithelial-mesenchymal transition (EMT), which is a crucial step in initiation of metastatic process. The authors found that inhibiting NF-kappaB signaling prevented EMT in Ras-transformed epithelial cells, while activation of this pathway promoted the transition to a mesenchymal phenotype. Furthermore, inhibition of NF-kappaB activity in mesenchymal cells caused a reversal of EMT, suggesting that NF-kappaB is essential for both the induction and maintenance of EMT. In accordance with this previous report, Dai et al. [62] observed that upregulation of miR-423 led to activation of the NF-kappaB signaling pathway by repressing TNIP2 expression, suggesting that miR-423 is a crucial factor that enhances breast cancer cell invasion. Therefore, miR-423 overexpression as a consequence of the presence of the rs6505162 allele A could trigger an initiation of the metastatic process by activating the NF-kappaB signaling pathway. This finding highlights this miRNA as a promising prognostic and therapeutic marker for metastatic BC.”

d) Information regarding the potential use of rs6505162 in clinical practice has been added to the Abstract and Discussion section.

Reviewer 2 Report

In the article entitled “Genetic variation in microRNA-423 promotes proliferation, migration, invasion, and chemoresistance in breast cancer cells” Sebastian Morales-Pison and colleagues performed an in vitro evaluation of the role of the SNP rs6505162:C>A in breast cancer cell lines.

They suggest that rs6505162:C>A is a functional SNP site with potential utility as a marker for diagnosis, prognosis, and treatment efficacy in BC patients. Although presenting interesting data in cell lines that might be worth publishing, there are some points that should be clarified. Therefore, the article is not suitable for publication unless amendments are made.

Major point:

Results

Lines 106-107

“These results indicate that pre-miR-423-A significantly enhanced proliferation of both luminal A and triple-negative representative BC cells”. Did the authors tested the proliferation index of normal breast MCF-10A cell line which were transfected with their expression vectors containing pre-miR-423-A or pre-miR-423-C? Results should be presented.

Minor point:

Figures appear grainy. Please, improve the low quality and definition.

Author Response

Regarding the comments of reviewer 2:

a) The proliferation experiment was performed on the normal mammary epithelium cell line MCF-10A. This cell line was transfected with expression vectors containing pre-miR-423-A or the pre-miR-423-C. The results were added to the text in the Results section.

b) The quality and definition of the figures was improved.

Round 2

Reviewer 2 Report

The authors did amendments to the manuscript. Major and minor points were all addressed in the text. 

Thus, I recommend the article for publication.